# Neighbourhood walkability, leisure-time and transport-related physical activity in a mixed urban–rural area

Eric de Sa and Chris I. Ardern

School of Kinesiology and Health Science, York University, Toronto, Ontario, Canada

## ABSTRACT

**Objectives.** To develop a walkability index specific to mixed rural/suburban areas, and to explore the relationship between walkability scores and leisure time physical activity.

**Methods.** Respondents were geocoded with 500 m and 1,000 m buffer zones around each address. A walkability index was derived from intersections, residential density, and land-use mix according to built environment measures. Multivariable logistic regression models were used to quantify the association between the index and physical activity levels. Analyses used cross-sectional data from the 2007–2008 Canadian Community Health Survey ($n = 1158$; $\geq 18$ y).

**Results.** Respondents living in highly walkable 500 m buffer zones (upper quartiles of the walkability index) were more likely to walk or cycle for leisure than those living in low-walkable buffer zones (quartile 1). When a 1,000 m buffer zone was applied, respondents in more walkable neighbourhoods were more likely to walk or cycle for both leisure-time and transport-related purposes.

**Conclusion.** Developing a walkability index can assist in exploring the associations between measures of the built environment and physical activity to prioritize neighborhood change.

## INTRODUCTION

Despite the well-known benefits of physical activity (PA), less than half of the Canadian population achieves the recommended level of daily activity (*Statistics Canada, 2009*). While social support for exercise (*Sallis, 2009*) and other psychosocial and demographic factors are important determinants of PA, the neighbourhood environment may provide additional benefits to population health by enabling social modeling and removing environmental constraints. The potential effects that urban form may have on PA participation are different from the effects found with behaviour modification programs or individually-tailored interventions (*Saelens et al., 2003*). Instead of selecting small portions of the population who are motivated enough to volunteer and participate in different PA programs, changes in the built environment can potentially impact the entire population. Indeed, even modest associations between PA (e.g., walking and cycling for leisure or transport)

Corresponding author
Chris I. Ardern, cardern@yorku.ca

and the built environment could have a lasting impact if the manner in which people interact with the neighbourhood landscape can be permanently altered (*Saelens et al., 2003*).

Accumulating even modest amounts of daily PA ($\geq$1.5 (kcal/kg/day; KKD)) is associated with a range of health benefits, and can be achieved with as little as 30 min of walking per day (*Cameron, Craig & Paolin, 2004*). While walking and bicycling are two of the most frequently reported forms of PA (*Russell & Craig, 1995*), only 7% of Canadians walk or cycle to work (*Statistics Canada, 2006*). Walking and bicycling are also more frequently reported by those with higher levels of income (*Cameron, Craig & Paolin, 2004*), suggesting that any research on the determinants of walking and cycling behaviour must account for the local neighbourhood as an important moderator of short bouts of discretionary PA participation.

Previous research has typically used either geographic analyses to study overweight/obesity (*Pouliou & Elliott, 2009*; *Slater et al., 2009*), or hierarchical linear modeling (*Wendel-Vos et al., 2004*; *Li et al., 2005*; *Nagel et al., 2008*) approaches for the study of neighbourhood effects on PA participation. However, since many measures of the urban form are correlated, a third approach to avoid the problem of spatial multicollinearity has been proposed; to develop a walkability index that integrates various measures of the built environment (*Frank et al., 2005*; *Li et al., 2009*; *Frank et al., 2009*). The aim of the present study is to therefore develop a walkability index and apply it to respondents found within York Region, Ontario, to explore the associations between this index with patterns of PA participation using health survey data.

## METHODS

### The Regional Municipality of York (York Region)

York Region is located directly north of Toronto and comprises nine municipalities: City of Markham, City of Vaughan, Town of Richmond Hill, Town of Aurora, Town of Newmarket, Township of King, Town of Whitchurch-Stouffville, Town of East Gwillimbury, and Town of Georgina. Of these, the population growth rates are highest amongst the three municipalities that are closest to Toronto (Vaughan, Markham, and Richmond Hill). The 2006 census profile estimated the population to be 892,712 and increased to 1,032,524 in 2011 (*Statistics Canada, 2011*). During the period of 1996–2001, York Region was the fastest growing Census Division in Canada, 30% of whom identified themselves as visible minorities (*York Region, 2003*). By 2010, the total population had exceeded one million people, and from 1996 to 2001, there was a 30% increase in the employment labour force (from 297,600 to 387,700), and this number is projected to increase to 800,000 jobs by 2031 (*York Region, 2009*). As a result, this population offers a unique opportunity to look at a demographically diverse, semi-rural/suburban region, that is characteristic of new growth in regions surrounding major municipalities.

### Canadian Community Health Survey (CCHS) 2007–2008

This analysis uses data from the 2007 to 2008 Canadian Community Health Survey (CCHS 2007–2008, master data file; Statistics Canada, Health Statistics Division and

Special Surveys Division), obtained through the limited data access program at the York University chapter of the Toronto Research Data Center of Statistics Canada. The CCHS is a cross-sectional survey that collects information on health status, health care utilization, and health determinants. To give equal importance to the health regions in each province, a multi-stage sample allocation strategy was employed.

The CCHS questions are designed for computer-assisted interviewing (CAI). Approximately 130,000 persons across 121 health regions were sampled during the data collection period from January 2007 to December 2008 inclusive. Three sampling frames were used to select the sample of households: 49% of respondents were obtained from an area frame, 50% from a list frame of telephone numbers, and the remaining 1% from random digit dialing. Interviews were conducted both in person and over the telephone. Some editing of the data was performed at the time of the interview by the interviewer using the CAI application. It was not possible for interviewers to enter out-of-range values, and flow errors were controlled through programmed skip patterns.

### Exclusion criteria

All Canadians aged 12 years and older were considered eligible for participation in the CCHS study (with few exceptions including individuals living on Indian Reserves or Crown Lands, institutional residents, full-time members of the Canadian Forces, and residents of certain remote regions). All respondents who were unable to be properly geo-coded with their corresponding postal-code address or whose address fell outside the York Region boundary (online: http://ww6.yorkmaps.ca/YorkMaps/nindex.html) were eliminated from analysis (final analytical sample = 1,158). For ease of interpretation, the present analysis was limited to respondents 18 years or older.

### Dependent variables (physical activity)

Respondents were asked frequency and duration of both walking and cycling for leisure (leisure-time physical activity) and to school/work (transport-related physical activity). Two dichotomous outcomes were derived from the indices: respondents were classified as having engaged in walking or cycling for leisure-time purposes (LPA: any/none) and walking or cycling for total PA (both leisure-time and transport-related purposes; TPA: any/none).

### Independent variables (built environment measures)

All built/neighborhood environment measures were quantified within a 500 m buffer zone and a 1,000 m buffer zone around the centroid of each postal code address. Buffer regions of 500 m and 1,000 m were chosen as they can be approximated to walking for 5 and 10 min, respectively (*US Department of Health and Human Services, 1996*; *Kondo et al., 2009*; *CFLRI, 2014*), and from a PA guideline perspective, even engaging in short sessions can help people work toward accumulating the minimum daily recommended levels of PA (*CSEP, 2010*). A measure of residential density was ascertained by calculating the number of dwellings (detached, semi-detached, condos, and apartments) and dividing by the total area of the buffer zone (units/hectare). Number of street intersections including those

with traffic lights and those without (excluding freeway ramps) were counted within each buffer zone. The algorithm for the evenness of distribution of square meters for each of the different land-use classifications was based on that of *Frank et al. (2005)*.

## Walkability index

Subsequently, two separate (i.e., 500 m and 1,000 m buffer zone) walkability indices were developed. A normalized distribution was taken for the residential density and the intersection variables (with removal of the lower and upper 5% measurements). To find the greatest explanatory power for variation in overall PA, a linear regression model for each built environment measure was analysed with a general measure of physical activity. Separate models were built by increasing the weight to the built environment measure (starting with no weight). When the variation accounted for by the model (with the weighted measure) did not increase between 2 models by more than 1%, no further weights were applied. Once this had been completed for each measure that comprised the walkability index, a new set of models were built combining 2 measures, each weighted to account for the greatest variation. Again, the weights were adjusted to account for the greatest explanatory variation before the third and final measure was added. These steps remain consistent with the approach used by *Frank et al. (2005)* as outlined in their original paper describing the model building and weighting process used to derive their walkability index. The end result for both the 500 m and 1,000 m walkability index is listed below:

*Walkability Index* $= (3xz$-score of land-use mix$)$

$+ (z$-score of net residential density$) + (z$-score of intersection$)$.

A higher walkability index would indicate that the respondent lived within a buffer region that was more walkable (suggested by higher number of intersections, higher residential density, and/or greater degree of land-use mix classification). Scores were then categorized into quartiles so that the first (lowest) quartile represented respondents living in the least walkable neighborhoods, and the fourth contained respondents with the most walkable neighborhoods.

Increasing the land-use mix weight beyond a weight of 3 (while holding the other variables constant) only marginally ($<$1%) increased the amount of variation accounted for by the walkability index. By contrast, overall predictive ability of the model was altered with further adjustment for weights associated with other explanatory variables, and in some cases resulted in a decrease in the explained variance. Although an updated version of the walkability index incorporating a retail floor area ratio (calculated as the retail building floor area footprint divided by retail land floor area footprint) has been proposed by *Frank et al. (2009)*, as area retail establishment data was unavailable, this variable was not incorporated into the index in the current analysis.

## Geographic Information Systems (GIS) software and statistical analysis

ArcView GIS, version 9.3 software (ESRI, Redlands, California, 2005) was used to geocode participants by postal-code address to existing maps in the CanMap StreetFiles: Ontario

and Platinum Postal Code Suite (both are products from DMTISpatial corporation) (DMTI Spatia, Inc. Markham, Ontario, Canada). The postal code polygons within the shape file differed in size depending on the area each represented. Most commonly, respondents are located on the periphery of each polygon; however, given that specific street and house/unit numbers were not available for the CCHS, respondents were geocoded to the centroid for these analyses. It is therefore expected that there would be a greater displacement from the periphery to the centroid for respondents belonging to postal code regions that cover larger (as compared to smaller) areas.

A series of map layers specific to each built environment measure (including: residential density, area of building space, area of parks/green spaces, and intersections) were used to quantify the characteristics within the 500 m buffer zone. The geocoding process resulted in the formation of a centroid to represent each 6-digit postal code region. Once data relating to the built environment measures were collated for each participant, the spatial data was quantified and exported into a SAS compatible database that was linked with the PA and individual-level covariates for each participant.

The walkability index was applied to a 500 m and 1,000 m buffer zone surrounding each respondent's 6-digit postal code address. Logistic regression was used to estimate the odds (OR, 95% confidence interval) of walking and/or cycling for both leisure-time and transport-related purposes across quartiles of the walkability index (quartile 1: OR = 1.00). Model 1 was the univariate association between the built environment measure and PA outcome and Model 2 adjusted for all other covariates in the multivariate model (age, sex, bmi, education, income, ethnicity, and smoking status) which have previously been shown to correlate with PA and cluster within neighborhoods (*Wendel-Vos et al., 2004*; *Frank et al., 2005*; *Nagel et al., 2008*; *Li et al., 2009*). Data analysis was conducted using SAS version 9.2 (Cary, NC) and statistical significance was set at alpha <0.05.

## RESULTS

Correlation coefficients are presented in Table 1. Within 500 m and 1,000 m boundaries, significant correlations exist between each of the built environment measures that are used to calculate the walkability index. The characteristics of the local built environment around each respondent's place of residence for each of the buffer zones are found in Table 2.

When a 500 m buffer zone was used, compared to respondents who lived in areas with the lowest walkability index scores (quartile 1), those living in both the third and fourth quartiles were 55% more likely to walk or cycle for leisure (Q3, OR: 1.55 CI 95% [1.07–2.26]; Q4, OR: 1.55 CI 95% [1.07–2.25]). This effect was also found when applying a 1,000 m buffer zone, because respondents were more likely to engage in walking/cycling for leisure when they lived in the second (OR: 1.53 CI 95% [1.05–2.21]), third (OR: 1.50 CI 95% [1.04–2.16]), and fourth (OR: 1.72 CI 95% [1.18–2.50]) quartiles. By contrast, within a 500 m buffer zone, higher walkability scores were not associated with higher odds of walking/cycling for leisure or transportation purposes, whereas the extended 1,000 m buffer revealed that only those in the most walkable neighbourhoods (fourth quartile)

**Table 1** Pearson correlation coefficients among individual measures of the walkability index.

|  | Intersections | Residential density | Land-use mix |
|---|---|---|---|
| **Unadjusted** | | | |
| Intersections | 1 | 0.32[*] | 0.15[*] |
| Residential density | 0.48[*] | 1 | 0.23[*] |
| Land-use mix | 0.22[*] | 0.24[*] | 1 |
| **Adjusted[a]** | | | |
| Intersections | 1 | 0.35[*] | 0.16[*] |
| Residential density | 0.50[*] | 1 | 0.21[*] |
| Land-use mix | 0.24[*] | 0.26[*] | 1 |

Notes.

Correlations above diagonal are for 500 m buffer; below diagonal are for 1,000 m buffer.

[*] $p < 0.001$.

[a] Adjusted for age, sex and education.

**Table 2** Characteristics of local built environment around respondent's places of residence, York Region, Ontario.

| Built environment measure | 500 m buffer | 1,000 m buffer |
|---|---|---|
|  | **Mean (SD)** | **Mean (SD)** |
| Residential density (units/hectare[a]) | 6.8 (4.4) | 6.5 (3.9) |
| Land-use mix[b] | 0.4 (0.2) | 0.5 (0.2) |
| Walkability index | −0.1 (4.0) | −0.5 (4.7) |
|  | **Range** | **Range** |
| Intersections | 0–133 | 0–330 |
| Walkability index | −10.4–8.5 | −16.2–8.4 |

Notes.

[a] 1 hectare = 10,000 square metres.

[b] Land-use mix = (−1) × [(hectares of commercial/total hectares of land use) × ln (hectares of commercial/total hectares of land use) + (hectares of government and institutional/total hectares of land use) × ln (hectares of government and institutional/total hectares of land use) + (hectares of open area/total hectares of land use) × ln (hectares of open area/total hectares of land use) + (hectares of parks and recreation/total hectares of land use) × ln (hectares of parks and recreation/total hectares of land use) + (hectares of residential/total hectares of land use) × ln (hectares of residential/total hectares of land use) + (hectares of resource and industrial/total hectares of land use) × ln (hectares of resource and industrial/total hectares of land use) + (hectares of waterbody/total hectares of land use) × ln (hectares of waterbody/total hectares of land use)]/ln (7).

Total hectares of land use = ∑(commercial, government and institutional, open area, parks and recreation, residential, resource and industrial, waterbody.)

were more likely to engage in walking/cycling for leisure and transportation purposes (OR: 2.22 CI 95% [1.22–4.02]) (Table 3).

## DISCUSSION

In order to assess the relationship between PA participation and the built environment, a composite measure of its features may provide a clearer understanding than an assessment of individual parts. Previous research has identified at least five inter-related dimensions of the built environment: density and intensity of development, mix of land uses, connectivity of the street network, scale of streets, and aesthetic qualities of a place

**Table 3** Association of physical activity with walkability index quartiles in 500 m and 1,000 m buffer zones.

| Outcome | Walkability index quartiles | Model 1[a] OR | CI | Model 2[b] OR | CI |
|---|---|---|---|---|---|
| **500 m buffer zone** | | | | | |
| Leisure physical activity | Quartile 1 | 1.00 | (ref) | 1.00 | (ref) |
| (Leisure PA for walking/cycling) | Quartile 2 | 1.26 | [0.89–1.78] | 1.44 | [1.00–2.08] |
| | Quartile 3 | 1.34 | [0.94–1.91] | **1.55** | [1.07–2.26] |
| | Quartile 4 | 1.27 | [0.89–1.80] | **1.55** | [1.07–2.25] |
| Total physical activity | Quartile 1 | 1.00 | (ref) | 1.00 | (ref) |
| (Leisure- and transport-related | Quartile 2 | 1.17 | [0.66–2.07] | 1.06 | [0.58–1.93] |
| PA for walking/cycling) | Quartile 3 | 1.65 | [0.95–2.86] | 1.33 | [0.74–2.39] |
| | Quartile 4 | 1.59 | [0.91–2.79] | 1.73 | [0.96–3.15] |
| **1,000 m buffer zone** | | | | | |
| Leisure physical activity | Quartile 1 | 1.00 | (ref) | 1.00 | (ref) |
| (Leisure PA for walking/cycling) | Quartile 2 | 1.30 | [0.91–1.84] | **1.53** | [1.05–2.21] |
| | Quartile 3 | 1.31 | [0.92–1.86] | **1.50** | [1.04–2.16] |
| | Quartile 4 | **1.43** | [1.00–2.04] | **1.72** | [1.18–2.50] |
| Total physical activity | Quartile 1 | 1.00 | (ref) | 1.00 | (ref) |
| (Leisure- and transport-related | Quartile 2 | 1.51 | [0.84–2.73] | 1.33 | [0.72–2.48] |
| PA for walking/cycling) | Quartile 3 | 1.75 | [0.98–3.13] | 1.54 | [0.84–2.81] |
| | Quartile 4 | **2.17** | [1.23–3.81] | **2.22** | [1.22–4.02] |

**Notes.**

[a] Model 1: unadjusted.

[b] Model 2: adjusted for the following covariates: age, sex, bmi, ethnicity, education, income, smoking status.

(*Handy et al., 2002*). In the present study, the resulting walkability index extends from previous research utilizing available GIS data (i.e., population density, land use mix, and intersections) (*Frank et al., 2005*; *Frank et al., 2009*). Although early attempts to increase walking and cycling behavior were initiated by transportation and urban planners as a means to reduce pollution and traffic congestion (*Sallis et al., 2006*), greater interest in ecologic models of behaviours have raised awareness of how the built environment may impact health and PA more specifically (*Sallis, 2009*). While major urban cities are often at the centre of interventions to explore the associations of PA within a community, a rapidly growing and demographically distinct municipality such as York Region offers a unique opportunity to explore associations between PA and a broader array of built environment features on PA frequency and participation. Our main finding was that after adjusting for demographic and health behaviours, a moderately-strong association between neighborhood walkability and PA was observed within a 500 m and 1,000 m buffer region for walking/cycling for leisure-time purposes, and within a 1,000 m buffer region for walking/cycling for total physical activity. These results reinforce and extend previous findings that residents living in more walkable neighbourhoods are more likely to engage in lifestyle-related PA, including those in semi-rural and suburban areas.

Despite the potential importance of these findings, it remains unclear if the respondents are substituting this neighbourhood-related PA for other more vigorous activity pursuits, or supplementing their existing PA because of the inherent walkability of their neighbourhood. This ambiguity is further compounded by not knowing the reason for each respondent's choice to live in their current neighborhood. If the respondent specifically chose their neighbourhood because of opportunities to engage (or not engage) in a more physically active lifestyle, they may be self-selecting an environment more conducive to continuing their pre-existing lifestyle. Nonetheless, if increases in walking were affected by the built environment, research suggests that even modest changes in walking time can have an impact optimally seen at the population level (*Nagel et al., 2008*). As an illustration, researchers suggested that an increase of 30 min of walking time per week equated to nearly 25% increase in the mean walking time found within their study sample, a shift that would result in nearly 30% of the sample meeting U.S. PA recommendations (*Nagel et al., 2008*).

Many built environment measures that are associated with increased PA behaviours (such as higher density of residential units and intersections) that are found within major urban environments may also act as reasons for why people may prefer to move away from urban centres and live in neighborhoods with a lower density of these features. As of 2007, there were approximately 485,000 people who worked in York Region with an estimated increase in jobs to 800,000 by 2031. Additionally, over one million people live within the municipality, with more than one-in-five projected to be over the age of 65 by 2026 (*York Region, 2009*). Irrespective of the type of neighborhood development, the large population base and workforce residing within York Region help illustrate the potential impact that these built environment measures could have on PA behaviours. Therefore, it is possible that even modest changes to neighborhood design might encourage increased walking and cycling behaviour and may have an overall effect on population PA levels.

## STRENGTHS AND LIMITATIONS

Despite the importance of these findings, there are a number of inherent limitations. First, developing an address locator to geocode the exact address for each respondent was not possible since data collected from the CCHS master file only contains the 6-digit postal code and not street name or numerical house/apartment location. Because the centroid of the polygon was used to map each respondent, using a 500 m or 1,000 m network buffer would likely increase estimation error since a more precise starting point would be needed to calculate network distance. A buffer based on Euclidean distance, while not directly related to the distance a respondent may travel to, still illustrates the local built environment features that makeup the respondent's neighborhood. Second, while previous studies have assessed the association between PA behaviours and the built environment using objectively measured accelerometery data, the variables captured within the CCHS are all self-reported and subject to healthy responder bias. Nonetheless, the observed associations suggest that even without objective measures, large datasets can still be used to differentiate PA behaviours between high and low walkable areas. Associations between perceived neighborhood safety and physical activity have been identified in previous

research as notable barriers to PA participation (*Humpel et al., 2004*; *Pikora et al., 2006*; *Bennett et al., 2007*; *Tucker-Seeley et al., 2009*). Perceptions about safety could not be assessed within these analyses as questions asked during the CCHS interview do not address neighborhood safety. In addition, as with all survey data, social desirability bias should be acknowledged. Lastly, data relating to building class was unavailable, meaning that the walkability index did not differentiate between retail, industrial, or other types of buildings. Specific information relating to the type of building could indicate for what reasons people would travel to and utilize the floor space, as retail malls and grocery stores would be accessible to the general public and may be frequented often, whereas office buildings would restrict who may access the floor space as well as the time of day and period during the week.

## CONCLUSION

This analysis continues a critical public health discussion of the health-environment interaction related to physical activity. Results of the present study suggest that participants living in neighborhoods with the highest scores on the walkability index are more likely to engage in LPA and TPA. Because many measures of urban form consider environmental features that are in close proximity, creation of a walkability index may be necessary to avoid multicollinearity. In addition, the index may assist researchers in identifying regions within a neighborhood that are conducive to PA, and conversely, to help identify regions of inactivity that could eventually be used to target regions for intervention. The current analyses demonstrate that a widely used nationally representative health survey can be used to explore associations between captured physical activity and measures of the built environment.

### Funding

These analyses were unfunded.

### Competing Interests

The authors declare there are no competing interests.

### Author Contributions

- Eric de Sa conceived and designed the experiments, performed the experiments, analyzed the data, contributed reagents/materials/analysis tools, wrote the paper, prepared figures and/or tables, reviewed drafts of the paper.
- Chris I. Ardern conceived and designed the experiments, performed the experiments, contributed reagents/materials/analysis tools, wrote the paper, reviewed drafts of the paper.

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
