# Peer review of "Neighbourhood walkability, leisure-time and transport-related physical activity in a mixed urban–rural area"

_PeerJ, doi:10.7717/peerj.440_

## Round 0.1 · original submission · Major Revisions

In general the paper is well written and addresses an important topic. The key issue raised by the reviewer #2 - which is never easy to address - involves the dependence of the measures that characterize participants. If feasible the suggested statistical methods would be of value. In addition the proposal that some graphic material - such as maps - be added would be useful, assuming they are available.

Reviewer 1 ·

Basic reporting

- Please consider changing the order of “METHODS” in order to make it easier to read. I suggest starting with a small paragraph explaining all that is going to be explained in Methods. Then, I will start with a setting of the study area, then explaining the CCHS, exclusion criteria, variables and the rest in the same order that is now in the text.

Experimental design

- As you are comparing your index with walking and cycling behavior, it is better to change the name of “walkability” index (your purpose is not only to explore the features related to walking and walking behavior), to “physical activity environment” index.

- Please include the reason why you think your research is important.

Validity of the findings

No comments

Additional comments

After reading this current Manuscript I would like to make several General and Specific comments.

General Comments

- Please change the title, including something more specific. I suggest including the study area, the type of urban environment (suburban) or that you are going to develop a PA environment index and then use it.
- Please consider including some maps of the study area.

Specific Comments

Methods
L70: Please rephrase, It is not clear what do you try to say with “secondary analysis”

L92: please include the total population of the area.

L110: please provide another name for LTPA combined (e.g. PA combined), it may confuse with LTPA.

Results
L206: please include “both” third and fourth quartiles. If not, it appears that is third+fourth quartiles.

Discussión
L250-252: please rephrase

Conclusion
L288: please rephrase

·

Basic reporting

- The introduction is succint and fine. Authors may consider mentioning previous works on population level changes in physical activity and Geoffrey Rose's ideas in the last part of the first paragraph
- The paper could very much use a map with the study area. Plotting the walkability index may provide the reader with an overview of how the area looks in terms f exposure.
- Table 1 could be more informative. Maybe classify by exposure or outcome (like any/none physical activity).
- Table 2 is quite redundant , some of its information is available in table 1. The authors should consider merging both tables.
- In table 3 LTPA and TRPA should be spelled out.

Experimental design

- The outcome variable (leisure and transport related physical activity) looks too insensitive. The authors should consider using more granularity in the definition of their outcome (quartiles of physical activity?). It is unclear whether the first mentioned definition (inactive, moderately active and sufficiently acftive) is used anywhere in the analyses. If it is not it may be removed for clarity.
- The definition of the exposure is well detailed and allows reproducing the index.
- The main statistical model used by the authors (logistic regression) may fail to take into account a key characteristic of the population of this study: the assumption of independence of observations is hard to justify in a neighborhood study. Individuals within zipcodes are not independent of each other (same exposure, other factors affecting outcome all together) , therefore violating assumptions of independence of observations. The authors should consider using models that account for this lack of independence (either robust calculations of standard errors taking clustering into account or multilevel models). If the authors decide to use the latter, a detailed explanation of the model would be in order.
- Regarding the choice for covariates for adjustement, authors should consider creating an intermediate model only adjusted for age and sex as basic confounding variables. Authors should explain (probably in the discussion) whether adjusting by education/income/ethnicity would not remove the effect of these association given that they may be affected by the exposure (a given neighborhood affects people's opportunities and may be a cause for segregation). I would suggest having a model 1 (unadjusted), model 2 ( age and sex adjusted) and model 3 (adjusted for all mentioned variables)

Validity of the findings

- A large source of confounding for these kind of findings would be self-selection of active individuals into walkable neighborhoods. The authors mention this phenomenon in the discussion but it should be further ellaborated, maybe outlining how this study IS or IS NOT affected by self-selection. This should be incorporated into the limitations part of the discusison.
- Authors, coherently with their deisng, state the importance of small changes in physical activity. They may consider discussing the body of literature incorporating these kind of exposre measurements into sports-related physical activity and how their results add to this body of literature.
- Authors mention the lack of safety data. They should consider ellaborating more on what is meant by safety in this context.
- The last part of the conclussion ("hot spots for inacitivty") highlights an interesting (and novel) concept that is not developed in other parts of the manuscript. Authors may consider ellaborating more on it.

Additional comments

This paper is very straight forward. The major drawback is the assumption of independence of observations and the lack of a statistical model that controls for this.

---

## Round 0.2 · accepted · Accept

Thank you for the thorough and useful response to the reviewers comments and suggestions. The article is now accepted for publication.